# The Cancer Chemopreventive and Therapeutic Potential of Tetrahydrocurcumin

**DOI:** 10.3390/biom10060831

**Published:** 2020-05-29

**Authors:** Ching-Shu Lai, Chi-Tang Ho, Min-Hsiung Pan

**Affiliations:** 1Department of Seafood Science, National Kaohsiung University of Science and Technology, Kaohsiung 811, Taiwan; chlai@nkust.edu.tw; 2Department of Food Science, Rutgers University, New Brunswick, NJ 08901, USA; ctho@sebs.rutgers.edu; 3Institute of Food Science and Technology, National Taiwan University, Taipei 10617, Taiwan; 4Department of Medical Research, China Medical University Hospital, China Medical University, Taichung 40402, Taiwan; 5Department of Health and Nutrition Biotechnology, Asia University, Taichung 41354, Taiwan

**Keywords:** cancer chemoprevention, tetrahydrocurcumin

## Abstract

In recent decades, cancer has been one of the leading causes of death worldwide. Despite advances in understanding the molecular basis of tumorigenesis, diagnosis, and clinical therapies, the discovery and development of effective drugs is an active and vital field in cancer research. Tetrahydrocurcumin is a major curcuminoid metabolite of curcumin, naturally occurring in turmeric. The interest in tetrahydrocurcumin research is increasing because it is superior to curcumin in its solubility in water, chemical stability, bioavailability, and anti-oxidative activity. Many in vitro and in vivo studies have revealed that tetrahydrocurcumin exerts anti-cancer effects through various mechanisms, including modulation of oxidative stress, xenobiotic detoxification, inflammation, proliferation, metastasis, programmed cell death, and immunity. Despite the pharmacological similarities between tetrahydrocurcumin and curcumin, the structure of tetrahydrocurcumin determines its distinct and specific molecular mechanism, thus making it a potential candidate for the prevention and treatment of cancers. However, the utility of tetrahydrocurcumin is yet to be evaluated as only limited pharmacokinetic and oral bioavailability studies have been performed. This review summarizes research on the anti-cancer properties of tetrahydrocurcumin and describes its mechanisms of action.

## 1. Introduction

Turmeric, a dried, orange-yellow-colored rhizome derived from *Curcuma longa* L., belongs to the ginger family (Zingiberaceae) and is used as a spice and food additive. It has also been used in China and India for thousands of years as a traditional medicine to treat various diseases, such as inflammatory conditions, hepatic and gastric disorders, infections, and coughs [1]. Furthermore, in the past 50 years, the biological properties of turmeric have been extensively and scientifically demonstrated. Curcumin, an active ingredient of turmeric, was identified by Vogel and Pelletier in 1815 [2]; it was first reported in 1949 to have antibacterial activity [3]. Biological investigation of curcumin has increased substantially since the 1990s [4]. Numerous in vitro, in vivo, and clinical studies have examined the benefits of curcumin in the prevention and treatment of diseases. In parallel, interest in the chemical composition of turmeric has increased dramatically [5]. Over 230 phytochemicals have been isolated and identified from turmeric, including terpenoids, curcuminoids, flavonoids, steroids, alkaloids, and other phenolic compounds. These compounds, together with curcumin, contribute to the wide spectrum of biological activities of turmeric and are an important source for the development of various nutraceuticals and pharmaceuticals [6].

Curcuminoids, including curcumin, demethoxycurcumin, and bisdemethoxycurcumin, account for 2–9% of the active compounds of turmeric, depending on the species and the cultivation and processing conditions of the rhizome [5]. They belong to the group of diarylheptanoids with an aryl-C7-aryl skeleton and are considered the principal active constituents responsible for the biological functions of turmeric [7]. Among them, curcumin is the most abundant and has been extensively investigated, with more than 9000 published articles listed in the PubMed database. Several studies have highlighted curcumin’s broad range of physiological and biological properties that may aid in the prevention and management of diseases such as cardiovascular and liver diseases, hypertension, obesity, diabetes, neurological disorders, inflammation, skin diseases, fibrosis, and arthritis. The underlying mechanisms include modulation of various signaling molecules, transcription factors, and enzymes and regulation of epigenetic activity [8,9]. Curcumin has also received much attention for its chemopreventive and therapeutic potential against cancer, well documented by a comprehensive cellular and animal study [10,11]. The anti-cancer mechanism of curcumin has been proposed to be attributable to its anti-oxidant, anti-proliferation, anti-inflammation, anti-invasion, anti-metastasis, and anti-angiogenesis properties as well as its ability to induce cell-cycle arrest and programmed cell death (apoptosis and autophagy) [8,10]. Moreover, several clinical studies have evaluated the efficacy of curcumin against pancreatic, colon, breast, and prostate cancer, either alone or in combination with traditional chemo- and radiotherapy [11].

Although the potent anti-cancer effect of curcumin has been widely reported, its clinical applications are limited due to poor systemic bioavailability resulting from chemical instability, poor absorption, rapid metabolism, and rapid elimination [12]. Several promising approaches have been used to resolve the limited absorption of curcumin, including nanotechnologies such as polymeric nanoparticles, liposomes, nanogels, peptide and protein formulations, and cyclodextrin complexes [13]. Piperine augments curcumin bioavailability by increasing the absorption and inhibiting glucuronidation [14]. Furthermore, structural modification of curcumin improves its chemical stability and chemical and pharmacokinetic properties [15].

In the past decade, to improve the kinetic stability of curcumin, metabolites, and degradation, products of curcumin metabolism have been identified and isolated. This could help to understand curcumin biotransformation and assist in the identification of unique compounds derived from it. The poor bioavailability of curcumin is contradictory to its multiple biological efficacies [16]. After oral administration, curcumin undergoes biotransformation through reduction and glucuronidation in vivo, thereby producing diverse metabolites, several of which have shown more chemical stability than the parent molecule. As a result, it has been suggested that degradation products and metabolites of curcumin are responsible for its various biological activities [17]. The different chemical structures of metabolites may have physiological properties and molecular targets distinct from that of curcumin; thus, additional scientific studies are necessary to identify additional potential health benefits.

Tetrahydrocurcumin is of special interest because it displays better bioavailability and different biological activity and molecular mechanisms than curcumin. Tetrahydrocurcumin is well known for its anti-oxidative, anti-diabetic, and cytoprotective activities, reviewed previously [18,19]. Studies have also suggested that tetrahydrocurcumin is more potent than curcumin in the prevention and treatment of various cancers (as discussed below in Section 3). Although tetrahydrocurcumin has been studied for its antioxidative potential, its contribution to the known pharmacology of curcumin has not received sufficient attention. This review summarizes the anti-cancer properties of tetrahydrocurcumin and explores its potential mechanisms of action.

## 2. Tetrahydrocurcumin, a Candidate Metabolite of Curcumin

Many studies have been performed to clarify the pharmacokinetic parameters and metabolism of curcumin. The maximum plasma concentration (*C*_max_) and area under the curve (AUC) after oral administration were found to be significantly lower than those found after intravenous injection and indicate low absorption and first-pass liver metabolism after ingestion [20]. It has been suggested that after oral ingestion, curcumin undergoes rapid Phase I and Phase II metabolism mainly in the liver, intestines, and gut microbiota. Still, its metabolites have also been detected in other organs [17,21]. Reduction and conjugation reactions are the primary metabolic pathways of curcumin. In Phase I, alcohol dehydrogenase catalyzes the reduction of the double bond of curcumin to form dihydrocurcumin, tetrahydrocurcumin, hexahydrocurcumin, and octahydrocurcumin. Phase II metabolism occurs extensively in the liver and intestine via conjugation of glucuronide and sulfate by glucuronidase and sulfotransferase, respectively, to the phenolic positions of curcumin [17]. Glucuronidation is suggested to be the predominant pathway of conjugation. Curcumin and the reduced metabolites are also subjected to Phase II metabolism into glucuronide and sulfate conjugates [22]. Tetrahydrocurcumin-glucuronide and hexahydrocurcumin-glucuronide are major biliary metabolites, and dihydroferulic acid and ferulic acid are minor metabolites [23]. Curcumin also undergoes intestinal microbial metabolism (e.g., by *Escherichia coli*), by a two-step reduction from the conversion of curcumin to dihydrocurcumin and subsequently to tetrahydrocurcumin by nicotinamide adenine dinucleotide phosphate hydrogen (NADPH)-dependent reductase [24] (Figure 1). In addition, studies have suggested that other metabolites are also generated via dehydroxylation, cyclization, and methylation [25].

### 2.1. Characterization of Tetrahydrocurcumin

Tetrahydrocurcumin was first identified as a metabolite by Holder et al. in 1978 based on an in vivo metabolic study in rats using tritium-labeled curcumin [23]. The chemical structure of tetrahydrocurcumin is very similar to that of curcumin; however, it lacks the double bonds in the central seven-carbon chain within the molecule. The two conjugated double bonds in curcumin are responsible for the characteristic color of turmeric, while tetrahydrocurcumin is an off-white color due to the absence of the α, β-unsaturated carbonyl moiety. Studies have shown that tetrahydrocurcumin was isolated mainly in the conjugated forms in serum and various tissues such as the intestine, liver, kidneys, lung, muscle, pancreas, spleen, and brain [26,27,28]. Nevertheless, tetrahydrocurcumin in unconjugated and conjugated forms were identified in the liver and plasma of mice fed with curcumin [28]. Unconjugated tetrahydrocurcumin was also detected in the plasma and brain of mice treated with curcumin by various routes of administration and after a chronic dietary treatment [29].

Although tetrahydrocurcumin is considered a curcumin metabolite, it also naturally exists in the roots of *Zingiber mioga*, *Z. officinale*, and *Curcuma zedoaria* [30,31]. Moreover, tetrahydrocurcumin can be chemically synthesized through the hydrogenation of curcumin to reduce the double bonds [32]. The cultured cells of *Marchantia polymorpha* (common liverwort) were also found to convert curcumin into tetrahydrocurcumin, with a yield of 90% per day [33] (Figure 1). Studies have demonstrated that tetrahydrocurcumin displays better solubility and chemical stability than curcumin. Pan et al. reported that tetrahydrocurcumin is stable in 0.1 M phosphate buffers at a range of pH values. In addition, tetrahydrocurcumin is more stable than curcumin in phosphate buffer at physiological pH [22]. Furthermore, tetrahydrocurcumin is more stable than curcumin in aqueous solution at pH 8.0, with no detectable decomposition within 2 h [34]. Tetrahydrocurcumin is more stable and degrades more slowly than curcumin: its terminal half-life in cell culture medium and plasma is 813 and 232 min, respectively, compared to 186 and 111 min for curcumin [35]. 

The superior oral absorption of tetrahydrocurcumin has been documented in in vivo studies. After 4-week daily oral administration of tetrahydrocurcumin, larger levels of free tetrahydrocurcumin, and its conjugates (sulfates and glucuronides) were detected in the liver and serum than in mice fed curcumin [28]. Mice treated with tetrahydrocurcumin by oral gavage, intramuscular injection, and intraperitoneal injection showed a higher plasma level of free tetrahydrocurcumin by all administration routes than mice administered curcumin. A similar result was found in Tg2576 APPsw mice provided daily dietary administration of 0.5% tetrahydrocurcumin and curcumin for four months [29] These studies indicate that tetrahydrocurcumin exhibits better gastrointestinal (GI) absorption than curcumin; however, it may also be more resistant to hydrolysis in plasma. Nevertheless, the oral bioavailability of tetrahydrocurcumin is still considered poor in rats. This study also reported that tetrahydrocurcumin is primarily excreted via the non-renal route due to the low concentration of tetrahydrocurcumin found in urine [36]. 

### 2.2. Beneficial Effects of Tetrahydrocurcumin on Human Diseases

Tetrahydrocurcumin has many more desirable biological and pharmacological properties than curcumin. It has been recognized as a superior antioxidant that is responsible for its chemo-preventive efficacy against various diseases, including hypertension, atherosclerosis, diabetes, neurotoxicity, cardiovascular disease, hepatotoxicity, and liver fibrosis [18]. Curcumin also acts as an anti-oxidant; it may elicit pro-oxidant effects at higher dosages due to its phenyl hydroxyl groups, phenyl methoxy groups, and α, β-unsaturated carbonyl moiety, as well as the ability to modify enzyme activity that results in reactive oxygen species (ROS) formation [37,38,39,40]. In contrast to curcumin, tetrahydrocurcumin does not generate ROS because it lacks conjugated bonds in the central seven-carbon chain [39]. The anti-oxidative mechanism of tetrahydrocurcumin has been proposed to be due to its ability to scavenge radicals by phenyl hydroxyl groups; more importantly, though, its activity involves the cleavage of the C-C bond at the active methylene carbon between the two carbonyls in the β-diketone moiety during oxidation. This cleavage results in the production of o-methoxy phenol products that also act as antioxidants [41]. Another anti-oxidative mechanism of tetrahydrocurcumin appears to be its ability to increase the activities of anti-oxidative enzymes such as superoxide dismutase, catalase, glutathione peroxidase, and glutathione-S-transferase; tetrahydrocurcumin was shown to be more potent than curcumin in several corresponding in vivo studies [18,28,42,43]. 

It has been accepted that tetrahydrocurcumin is less cytotoxic than curcumin and displays cytoprotective properties against ROS-induced cellular damage and death [44,45,46]. Indeed, tetrahydrocurcumin has exhibited low toxicity in several in vivo studies [46,47]. In an acute toxicity study, 100–10,000 mg/kg tetrahydrocurcumin administered orally for 14 days caused no abnormal behavioral changes, convulsion, or death [48]. Tetrahydrocurcumin was also non-toxic in immunodeficient mice orally administered a 3000 mg/kg dose of tetrahydrocurcumin for 14 days [46]. In a 90 day sub-chronic toxicity study in rats, tetrahydrocurcumin at doses of up to 400 mg/kg showed no mortality and no reproductive or adverse effects [49]. Tetrahydrocurcumin has been shown to extend the lifespan of *Drosophila* [50]. In addition, in 13-month-old mice, long-term intake of 300 mg/kg tetrahydrocurcumin resulted in a longer average life span [47]. These studies show that tetrahydrocurcumin might be safe for pharmaceutical and nutraceutical applications. 

Another avenue by which tetrahydrocurcumin may be more beneficial than curcumin is its molecular targets, which are different from those of the parent molecule [19]. Despite the diketone moiety and phenolic groups being the major chemically reactive functional groups of curcumin, the α, β-unsaturated diketone moiety is considered the most important nucleophilic-addition reaction (known as the Michael addition) for interaction with cellular molecules, proteins, and enzymes. However, tetrahydrocurcumin is unable to act as an acceptor in the Michael reaction due to the absence of the unsaturated diketone and, hence, seems incapable of directly interacting with proteins and enzymes [19]. Nevertheless, tetrahydrocurcumin has exhibited its ability to inhibit enzyme activity in several in vitro and docking studies [36,51,52]. It is hypothesized that tetrahydrocurcumin is more molecularly flexible than curcumin due to the lack of double bonds in the hydrocarbon chain joining the aromatic rings of the molecule, thereby demonstrating better binding energy in interactions with enzymes [52]. Altogether, the above studies may support the notion that the biological potential of tetrahydrocurcumin arises from its distinct molecular mechanism and action on targets. 

## 3. Anti-Cancer Effects and the Underlying Mechanisms of Tetrahydrocurcumin Activity

Cancer is the second leading cause of morbidity and mortality worldwide. Tumorigenesis is a complicated pathological process involving both genetic and epigenetic alterations of gene expression (Figure 2). Dysregulation of gene expression drives various abnormalities in cellular functions. In 2000, Hanahan and Weinberg proposed six hallmarks of cancer: self-sufficiency in growth signals, insensitivity to anti-growth signals, limitless replicative potential, evading apoptosis, sustained angiogenesis, tissue invasion, and metastasis [53]. Two additional emerging hallmarks, reprogramming of energy metabolism and evasion of immune destruction, were subsequently added in 2011 [54]. These hallmarks have served as the principal foundation for our knowledge of multi-stage tumorigenesis. Although these concepts are widely accepted, understanding the molecular mechanisms of tumorigenesis and cancer progression remains elusive, and cancer continues to be associated with high morbidity and mortality. Recent research suggests that cancer development is a dynamic process [55]. The cancer cells communicate with each other and with stromal cells to create a microenvironment for their proliferation and growth while also continuing to evolve in response to diverse environmental factors. This evolving process may ultimately result in intra-tumor heterogeneity that is characterized by distinct tumor-cell populations with different phenotypic and molecular features within tumor tissue. There is growing evidence that heterogeneity is associated with therapeutic resistance, poor prognosis, and poor outcome [55]. Although cancer is still enigmatic to a degree, knowledge about tumorigenesis is accumulating, providing new information that serves in diagnosis and generation of novel therapeutic strategies against various human malignancies [56]. Curcumin is responsible for most of the anti-cancer effects of turmeric, but its poor bioavailability severely limits its pharmacological applications. Tetrahydrocurcumin shares structural features with curcumin but exhibits better solubility, chemical stability, and bioavailability. The resulting different chemical characteristics, biological properties, and distinct molecular targets of tetrahydrocurcumin suggest its potential application in cancer prevention and treatment. We summarize below the proposed tetrahydrocurcumin mechanisms of action against various malignancies (Table 1). 

### 3.1. Anti-Oxidative Activity

Oxidative stress, defined as an imbalance between pro-oxidants and antioxidants, is implicated in the pathogenesis of cancer. Excessive free radicals, ROS, and reactive nitrogen species (RNS) are known to participate in cancer initiation, promotion, and progression by influencing cell transformation, genomic instability, survival, proliferation, differentiation, metabolism, angiogenesis, and invasiveness. The concentrations of ROS are higher in cancer cells than in normal cells, which has a distinct impact on cellular functions [57]. ROS play an important role in the initiation of carcinogenesis, triggering gene mutation via DNA damage. This leads to the activation of proto-oncogenes or inactivation of tumor suppressor genes. ROS also attack other macromolecules such as proteins and lipids, affecting protein-enzyme function and causing lipid peroxidation. The resulting changes in the lipid bilayer of cell membranes lead to cell dysfunction [58]. 

**Table 1 biomolecules-10-00831-t001:** Proposed anti-cancer mechanisms and molecular targets of tetrahydrocurcumin.

	Targeted Cancer	Experimental Model	Tetrahydrocurcumin Concentration	Mechanism of Actions	References
Anti-oxidative activity					
Radical scavenging	-	DPPH radical solution	IC_50_ = 4.1 ~ 20.7 μM	Radical scavenging	[32,59,60]
	-	PMA-induced cells	IC_50_ = 200 μM	O_2_^-^• Radical scavenging	[60]
	-	Hypoxanthine/xanthine oxidase reaction	300 μM	O_2_^-^• Radical scavenging	[59]
	-	Fe_2_SO_4_/H_2_O_2_ reaction	200 μM	OH• Radical scavenging	[59]
	-	SNAP reaction	IC_50_ = 104.2 μM	NO Radical scavenging	[59]
Reduction of oxidative damage	-	AAPH-induced linoleic oxidation	1–12 μM	Reduced lipid peroxidation	[32]
	Renal	Fe-NTA-induced renal damage in male ddY mice	0.5% in diet for 4 weeks	Reduced DNA, lipid, and protein oxidative damage	[28]
Induction of antioxidant	Renal	Fe-NTA-induced renal damage in male ddY mice	0.5% in diet for 4 weeks	Upregulated antioxidant enzymes	[28]
	Liver	As-induced hepatotoxicity in male albino Wistar rats	80 mg/kg for 4 weeks	Reduced lipid peroxidation Upregulated antioxidant enzymes	[61]
	Liver	Cd-induced hepatotoxicity in male albino Wistar rats	20, 40, and 80 mg/kg for 4 weeks	Reduced lipid peroxidation Upregulated antioxidant enzymes	[62]
Modulation of Phase I and Phase II enzymes	-	Murine hepatoma cells Hepa 1c1c7	-	Upregulated NAD(P)H: quinone oxidoreductase	[63]
	-	Human CYP450 enzymes	0.01–100 μM	Inhibited CYP2C9 and CYP3A4 activity	[36]
	-	Acetaminophen-induced liver injury in male Kunming mice	25, 50, and 100 mg/kg	Downregulated *CYP2E1* gene Downregulated Kepa1 Upregulated Nrf2 and downstream genes	[64]
Anti-inflammatory activity	-	TPA-stimulated HL-60 cells and mouse skin	in vivo study: 810 nmol	Decreased ROS production Decreased MPO activity	[65]
	-	LPS-treated RAW264.7 macrophage and C57BL/6 mice	in vitro study: 10 and 50 μM in vivo study: 100 mg/kg	Decreased TNF-α production Downregulated phospho-IκB and NF-κB	[66]
	-	LPS-treated RAW264.7 macrophage	3.125–100 μM	Decreased NO, TNF-α, IL-6 production Downregulated NF-κB nuclear translocation	[67]
	Colon cancer	DSS-induced colitis in ICR mice	0.1 and 0.25 mmol/kg for 7 days	Downregulated NF-κB and STAT3 DNA binding activity Downregulated iNOS and COX-2	[68]
	-	Soybean lipooxygenase	1, 10, and 250 μg/mL	Downregulated lipoxygenase activity	[36]
	-	Molecular docking assay	-	Possible phospholipase A2 inhibitor	[52]
Anti-proliferative activity	-	TPA-stimulated mouse skin and JB6 cells	in vitro study: 5 and 10 μM in vivo study: 1 and 3 μmol	Reduced ODC activity Decreased anchorage-independent growth	[69]
	Colon cancer	DMH-initiated mice	Dietary 0.5% for 7 weeks	Reduced ACF formation and crypt proliferation	[70]
	Colon cancer	AOM-treated mice	Dietary 0.005 and 0.02% for 23 weeks	Reduced ACF formation Downregulated iNOS and COX-2 Decreased PCNA Downregulated Wnt-1/β-catenin/p-GSK3β signaling	[71]
	Glioma	Glioma cells (alone or combined with radiation)	3–161 μM	Reduced colony formation Induced G0/G1 cell cycle arrest Downregulated cyclin D1 and PCNA Increased GSH depletion	[72]
Anti-metastatic activity	Fibrosarcoma	HT1080 human fibrosarcoma cells	5–100 μM	Decreased invasion and migration Decreased cell adhesion Downregulated MMPs and uPA	[73]
	Liver cancer	HepG2 xenograft model	Oral 3000 mg/kg for 21 days	Decreased angiogenesis	[46]
	Cervical cancer	CaSki xenograft model	Oral 100, 300 and 500 mg/kg for 30 days	Decreased angiogenesis Decreased tumor volume Downregulated HIF-1α, VEGF, VEGFR2, EGFR, COX-2 Downregulated p-ERK1/2 and p-AKT	[74,75]
	Osteosarcoma	Lung metastasis model Osteosarcoma cell lines	in vitro study: 2.5–50 μM in vivo study: i.p. 100 mg/kg for 8 days	Reduced lung metastasis Decreased invasion and migration Promoted MET process Downregulated HIF-1α, VEGF, and MMPs Decreased hypoxia-induced angiogenesis	[76]
Induction of programmed cell death	Breast cancer	MCF-7 cells	15–130 μM	Induced mitochondria-dependent apoptosis Loss of ΔΨm Increased ROS production Upregulated Bax and Downregulated Bcl-2 Upregulated p21 Activation of caspases	[77,78]
	Liver cancer	H22 ascites tumor-bearing mouse model	i.p. 5, 10 and 20 mg/kg for 7 days	Induced mitochondria-dependent apoptosis Upregulated Bax and Downregulated Bcl-2 Activation of caspases Upregulated p53 and downregulated MDM2	[79]
	Leukemia	HL-60 cells	25–100 μM	Induced autophagy Upregulated LC3 I/II and beclin-1 Downregulated PI3K/Akt and mTOR/p70S6K signaling	[80]
	Leukemia	Cytarabine-resistance HL-60 cells	5–100 μM	Induced autophagy Upregulated p62 and beclin-1	[81]
	Lung cancer	A549 cells	30–130 μM	Induced autophagy Upregulated LC3 I/II and beclin-1 Downregulated PI3K/Akt/mTOR signaling	[82]
Immuno-modulating activity	-	RAW264.7 macrophages and LPS-stimulated mouse splenocytes	1–10 μg/mL	Increased phagocytosis Increased NK cells activity	[83,84]

AAPH, 2′-azobis(2-amidinopropane)hydrochloride; ACF, aberrant crypt foci; Akt, protein kinase B; AOM, azoxymethane; Bax, Bcl-2-associated X Protein; Bcl-2, B cell lymphoma 2; COX-2, cyclooxygenase 2; CYP, cytochrome P450 enzyme; DMH, 1,2-dimethylhydrazine; DPPH, 2,2-diphenyl-1-picrylhydrazyl; DSS, dextran sodium sulphate; EGFR, epidermal growth factor receptor; Fe_2_SO_4_, ferric sulfate; Fe-NTA, ferric nitrilotriacetate; GSH, glutathione; H_2_O_2_, hydrogen peroxide; HIF-1α, hypoxia-inducible factor 1α; IL-6, interleukin-6; iNOS, onducible nitric oxide synthase; IκB, inhibitor of kappa B; LC3 I/II, Protein light chain 3 I/II; LPS, lipopolysaccharide; MDM2, murine double minute 2; MET, mesenchymal epithelial transition; MMPs, matrix metalloproteinases; MPO, myeloperoxidase; mTOR, mammalian target of rapamycin; NADPH, nicotinamide adenine dinucleotide phosphate hydrogen; NF-κB, nuclear factor kappa B; NO, nitric oxide; Nrf2, nuclear factor erythroid 2-related factor 2; O_2_^-^•, superoxide anion radical; ODC, ornithine decarboxylase; OH•, hydroxyl radicals; p70S6K, 70-kDa ribosomal S6 kinase; p-AKT, phosphorylated protein kinase B; PCNA, proliferation cell nuclear antigen; p-ERK1/2, phosphorylated extracellular signal-regulated kinases1/2; p-GSK3β, phosphorylated glycogen synthase kinase 3β; PI3K, phosphatidylinositol-3-kinase; PMA, phorbol 12-myristate 13-acetate; ROS, reactive oxygen species; SNAP, S-nitroso-*N*-acetylpenicillamine; STAT3, signal transducer and activator of transcription -3; TNF-α, tumor necrosis factor α; TPA, tetradecanoylphorbol-13-acetate; uPA, urokinase-type plasminogen activator; VEGF, vascular endothelial growth factor; VEGFR2, vascular endothelial growth factor receptor 2; Wnt-1, Wnt family member 1; ΔΨm, mitochondrial transmembrane potential.

Tetrahydrocurcumin is a proven, potent anti-oxidant that has been shown to scavenge free radicals, decrease ROS production, and upregulate antioxidant enzymes and thus reduces oxidative stress and pathological conditions in tissues such as liver, kidneys, brain, and blood vessels [18]. Many in vivo studies have demonstrated that tetrahydrocurcumin’s potential activity against various toxicants or carcinogen-mediated oxidative damage and cellular dysfunction is superior to that of curcumin [18,19]. Here, we focus on the anti-oxidative effects of tetrahydrocurcumin during tumorigenesis. 

Tetrahydrocurcumin has demonstrated the ability to scavenge radicals. In one DPPH-radical scavenging experiment, tetrahydrocurcumin displayed a better scavenging effect than curcumin [32,59], while another study showed that their scavenging activities were similar [60]. The reported DPPH radical-scavenging IC_50_ of tetrahydrocurcumin ranged from 4.1. to 20.7 μM; the differences may be attributed to experimental conditions such as concentrations of the radicals, incubation time, solvent type, and the type of measurement used. Similar results were also observed for superoxide anion (O_2_^-^•)-radical scavenging. Tetrahydrocurcumin showed better superoxide-anion-radical-scavenging activity (IC_50_: 200 μM) than curcumin (IC_50_ >1357 μM) in a phorbol 12-myristate 13-acetate (PMA)-stimulated cell suspension [60]. Another study demonstrated that tetrahydrocurcumin showed a lower potential for superoxide-anion-radical scavenging than curcumin in an enzymatic reaction of hypoxanthine and xanthine oxidase [59]. In addition, tetrahydrocurcumin was more potent than curcumin in scavenging hydroxyl radicals (OH•) in the Fenton reaction; however, it showed a scavenging IC_50_ comparable to curcumin (104.2 μM and 100.4 μM for tetrahydrocurcumin and curcumin, respectively) for *S*-nitroso-*N*-acetyl penicillamine (SNAP)-generated nitric oxide (NO) radicals [59]. The comparable radical-scavenging activities of tetrahydrocurcumin and curcumin were concluded to be independent of the lack of conjugated double bonds. Still, they may primarily be related to the structural characteristics of both compounds, including phenolic hydroxy groups, methoxy groups, and keto-enol moiety. The hydrogen-donating ability of tetrahydrocurcumin, derived from hydrogenation of the two conjugated double bonds in the structure, has been suggested to play an important role in its potent and higher radical-scavenging capacity than curcumin [41,85]. In addition, the hydroxyl-radical-scavenging activity of tetrahydrocurcumin in the Fenton reaction may not only contribute to its direct scavenging effect but may also affect iron chelation [85]. However, further investigations of the link between molecular structure and radical-scavenging capability are required. 

Tetrahydrocurcumin was also more effective in significantly reducing lipid peroxidation in 2,2′-azobis(2-amidinopropane) dihydrochloride (AAPH)-induced linoleic-oxidation reaction [32]. The C-C bond cleavage at the active methylene carbon of the β-diketone moiety and o-methoxyphenol derivative has been attributed to the anti-oxidative effect of tetrahydrocurcumin during lipid peroxidation [41]. It was confirmed in in vivo studies that tetrahydrocurcumin alleviated macromolecular oxidative damage. Ferric nitrilotriacetate (Fe-NTA) is known as a carcinogen that induces renal-cell carcinoma via oxidative damage and nephrotoxicity. Mice fed a 0.5% tetrahydrocurcumin-containing diet for 4 weeks showed a marked reduction of Fe-NTA-induced formation of DNA and protein adducts as well as lipid peroxidation in the kidneys. The renal concentrations of hydroxy-2′-deoxyguanosine (8-OHdG)- and 4-hydroxynonenal (HNE)-modified proteins were significantly lower following tetrahydrocurcumin treatment than curcumin treatment [28]. In the same study, mice fed tetrahydrocurcumin induced a greater activity of glutathione peroxidase (GPx), glutathione S-transferase (GST), and NADPH quinone reductase in the kidneys than that exhibited by curcumin. This could indicate that another potential mechanism contributing to the oxidative-damage protection by tetrahydrocurcumin appears to be the induction of antioxidant enzymes. Tetrahydrocurcumin-induced renal GST activity was more potent than that in the control group, implying the specificity of tetrahydrocurcumin activity [28]. The liver is critical for metabolism and removal of toxins, heavy metals, and xenobiotics but is also susceptible to pathological damage by toxicants. Studies have shown that dietary tetrahydrocurcumin at 80 mg/kg for 4 weeks substantially mitigated arsenic- and cadmium-induced hepatotoxicity by reducing hepatic lipid peroxidation, upregulating glutathione (GSH), GPx, superoxide dismutase (SOD), and catalase [61,62]. However, these studies concluded that in addition to the upregulation of antioxidant enzymes, the scavenging activity of tetrahydrocurcumin (discussed earlier) could be another anti-oxidative mechanism for its hepatoprotective effect.

### 3.2. Modulation of Phase I and Phase II Enzymes

The metabolic biotransformation of chemicals or carcinogens to electrophilic intermediates by Phase I enzymes is known to directly damage DNA and promote gene mutations as well as modify other macromolecules; this is the primary step in the initiation phase of carcinogenesis [86]. In contrast, Phase II enzymes catalyze the conjugation reactions by adding hydrophilic groups to the original molecules or intermediates derived from Phase I enzymes, thus facilitating their elimination via bile or urine. In general, Phase II enzymatic conjugation serves as a detoxifying step for cellular defense against toxic chemicals, drugs, and carcinogens. Modulation of Phase I and Phase II metabolism to reduce the biotransformation of carcinogens or increase their elimination represents a strategy for cancer chemoprevention [87]. Curcumin has been shown to activate or inhibit enzymes by a Michael-addition reaction due to the unsaturated α, β-diketone group. While tetrahydrocurcumin lacks direct Michael-addition reactivity, it still elevated NAD(P)H quinone oxidoreductase in murine hepatoma cells [63]. In fact, tetrahydrocurcumin upregulation of Phase I and II enzymes in in vivo studies resulted in protective effects against heavy metal- and drug-induced hepatotoxicity [62,64]. 

Tetrahydrocurcumin appears to upregulate Phase I and II enzymes through both direct and indirect mechanisms. Dinkova-Kostova et al. reported that the β-diketone of tetrahydrocurcumin may be able to serve as a Michael-reaction acceptor because of keto-enol tautomerism, and thus can induce Phase II enzymes. This was further evidenced by a modified molecule, dibenzoyl propane, that lacks β-diketone and is inactive as a Phase II enzyme inducer [62]. Novaes et al. also proposed a similar mechanism for the inhibition by tetrahydrocurcumin of human Phase I drug-metabolizing cytochrome P450 (CYP450) enzymes, including CYP2C9 and CYP3A4. The two keto- and enol-tautomers of tetrahydrocurcumin were roughly equal in proportion to their chromatographic conditions [36]. They also hypothesized that tetrahydrocurcumin, together with curcumin, may undergo *O*-dealkylation by demethylation of the aromatic methoxy groups, and thus bind to CYP450 enzymes to facilitate inhibition [36]. A recent study showed that pretreatment with tetrahydrocurcumin reduced hepatic *CYP2E1* gene and protein expression in acetaminophen-induced liver injury in mice and exhibited a stronger reducing effect than curcumin [64]. Molecular docking studies further discovered that tetrahydrocurcumin formed hydrogen bonds with Arg344, Gly411, and Asn143 residues in the active site of CYP2E1, and Leu215, Asn219, and Ser366 residues interacted with curcumin [64]. Moreover, tetrahydrocurcumin upregulated detoxifying enzymes through a transcriptional mechanism. It reduced cytosolic levels of Kepa1 protein and increased nuclear levels of Nrf2 protein, resulting in the upregulation of transcription of detoxifying molecules, glutamate-cysteine ligase catalytic subunit (GCLC), glutamate-cysteine ligase modifier subunit (GCLM), and heme oxygenase-1 (HO-1). The upregulation of GCLM and HO-1 was more potent after treatment with tetrahydrocurcumin than curcumin. The suggested mechanism of action was that tetrahydrocurcumin activity resulted from its insertion into the pocket of Keap1 by the formation of a hydrogen bond with the Val465 and Ser508 residues of Keap1 and thus occupied a part of the binding site of Nrf2; this generated interference in the binding between Keap1 and Nrf2, ultimately allowing the activation of Nrf2 and downstream targets [64]. Again, curcumin formed hydrogen bonds with the Keap1 protein at completely different residues than tetrahydrocurcumin. The data showing that tetrahydrocurcumin exerted stronger hepatoprotection than curcumin against the effects caused by acetaminophen were explained by its great potential for suppression of CYP2E1 and upregulation of Nrf2 that not only reduced the metabolic transformation of acetaminophen but also augmented oxidative damage [64]. Tetrahydrocurcumin also restored hepatic GST in cadmium-exposed rats; however, the mechanism of action was not investigated [62]. 

### 3.3. Anti-Inflammatory Activity

Chronic inflammation is a fundamental hallmark of a wide range of diseases, including cancers. Unlike innate inflammation with a rapid and self-limiting characteristic, chronic inflammation represents a dysregulated, unresolved, and prolonged pathological process. It has been shown to participate in all stages of tumorigenesis [88]. Chronic inflammation caused by genetic alteration, infection, or injury triggers infiltration of immune/inflammatory cells that release various oxidants and proinflammatory molecules, inducing genomic instability that leads to DNA damage and mutation of initiated cells. The various cytokines, chemokines, and growth factors produced by immune/inflammatory and tumor cells promote malignant transformation and proliferation of initiated cells, resistance to cell death, angiogenesis, invasion, and metastasis. During tumor progression, the coordination of proinflammatory responses via autocrine and paracrine patterns among cancer cells, stromal cells, and inflammatory cells maintain an inflammatory microenvironment for tumor growth [88,89]. Targeting of inflammation is believed to be a relevant strategy for cancer prevention and treatment. It can be achieved through the downregulation of inflammatory mediators, modulation of inflammatory signaling, suppression of inflammatory infiltration, and interference in the interaction between cancer and immune/inflammatory cells [89]. 

Several studies have shown that the anti-inflammatory effect of tetrahydrocurcumin was lower than that of curcumin. Huang et al. reported that tetrahydrocurcumin showed a maximum 50% inhibitory effect on the tumor promoter 12-O-tetradecanoylphorbol-13-acetate (TPA)-induced ear edema in female mice, while curcumin treatment achieved almost complete inhibition [69]. A similar effect was also observed in TPA-stimulated O_2_^-^• and H_2_O_2_ production in differentiated HL-60 cells and in mouse skin to which double-TPA had been applied. Tetrahydrocurcumin exhibited a weaker inhibition than that of curcumin on the recruitment and activity-suppression of inflammatory cells; this was demonstrated by the epidermal myeloperoxidase (MPO) activity that is highly expressed in activated-neutrophil granulocytes [65]. Tetrahydrocurcumin exerted anti-tumor-promotion activity in a 7- and 12-dimethylbenzanthracene/TPA-induced, mouse-skin tumorigenesis model, demonstrating a decreased tumor number and incidence even though it was less active than curcumin [69]. Several studies have suggested that tetrahydrocurcumin exerts relatively weak regulatory activity on inflammatory signaling and mediators compared to curcumin. In lipopolysaccharide (LPS)-stimulated murine macrophages and in mice in vivo, tetrahydrocurcumin was less effective at suppressing the phosphorylation of IκB, activation of nuclear factor-kappa B (NF-κB), and production of tumor necrosis factor-α (TNF-α) [66]. Surh et al. also demonstrated that tetrahydrocurcumin had a weaker inhibitory effect on the DNA binding activity of transcription factors, NF-κB, and signal transducer and activator of transcription 3 (STAT3) and downstream inflammatory enzymes, and consequently showed a weaker inhibitory effect than curcumin on dextran sulfate sodium (DSS)-induced mouse colitis [68]. However, Zhao et al. demonstrated that tetrahydrocurcumin suppressed LPS-induced nitric oxide (NO) production and TNF-α and IL-6 secretion in RAW264.7 murine macrophages and showed a comparable inhibitory effect to curcumin at higher concentrations. Tetrahydrocurcumin also effectively reduced the protein levels of inducible nitric oxide synthase (iNOS) and cyclooxygenase-2 (COX-2) as well as downregulated the nuclear translocation of NF-κB. However, the study did conclude that curcumin exerted a more potent inhibitory effect than tetrahydrocurcumin on the LPS-induced inflammatory response [67]. 

Many synthetic analogs of curcumin have been developed by modification of its structural motifs to exhibit greater chemical stability, bioavailability, and biological activities and have been discussed in several review articles [15,90,91]. Some of them displayed a better aqueous solubility, cellular uptake, in vivo bioavailability, and anti-cancer effect than original curcumin [92,93,94]; however, relatively few comparative studies were conducted between curcumin analogs and tetrahydrocurcumin. Pae et al. reported the dimethoxycurcumin, a synthetic analog with higher metabolic stability over curcumin, was more potent than curcumin and tetrahydrocurcumin to suppress NO production in LPS-stimulated RAW264.7 murine macrophages [95]. Their latter study also showed the synthetic dimethoxycurcumin was more effective than tetrahydrocurcumin on induction of Nrf-2 dependent HO-1 expression in RAW264.7 murine macrophages [96]. These studies indicated not only the two conjugated double bonds in curcumin are required for anti-inflammatory activity but also the increased number of methoxy groups.

The potency of the anti-inflammatory effect of tetrahydrocurcumin in LPS-stimulated macrophages may vary depending on the concentration of both tetrahydrocurcumin and LPS as well as the duration of treatment. A recent in vivo study reported that tetrahydrocurcumin had a superior anti-inflammatory effect than curcumin. Zhang et al. demonstrated that tetrahydrocurcumin at a lower dosage (40 mg/kg) exhibited a comparable and even stronger efficacy than curcumin (100 mg/kg) to reduce xylene-induced ear edema, carrageenan-induced paw edema, and acetic acid-induced vascular permeability in mice. They also found that tetrahydrocurcumin was more potent than curcumin in downregulating COX-2 protein in carrageenan-induced paw edema and indicated that tetrahydrocurcumin might be a better selective inhibitor than curcumin [48]. However, an in vitro COX-2 inhibition assay showed neither curcumin nor tetrahydrocurcumin to be an effective inhibitor of the COX-2 enzyme, while both displayed a potent inhibitory effect on lipoxygenase activity at a 1 μM concentration [36]. Most studies concluded that the weak anti-inflammatory activity of tetrahydrocurcumin was due to its inability to act as a Michael-reaction acceptor because of the lack of the α, β-unsaturated carbonyl moiety present in curcumin [19,36]. A molecular docking study showed that tetrahydrocurcumin possessed better binding energy than curcumin with phospholipase A2, an enzyme involved in the formation of many proinflammatory mediators, by forming a hydrogen bond at Gly30 residue at the active site (the bond is formed at Asp49 of curcumin). Tetrahydrocurcumin forms much stronger van der Waals bonds with the protein atoms of phospholipase A2 than curcumin. This study indicated that the loss of double bonds in the central seven-carbon chain of tetrahydrocurcumin may result in binding pattern differences from curcumin and consequently, a different regulating mechanism for inflammatory enzymes [52]. Overall, tetrahydrocurcumin appears to be very weak in modulating inflammatory signaling and mediators that are likely not responsible for any major mechanism of its anti-cancer activity. 

### 3.4. Anti-Proliferative Activity

Uncontrolled cell proliferation is a critical event in tumorigenesis and tumor expansion. Accumulation of gene mutations in initiated cells and constitutive activation of mitogenic signaling pathways alter the expression and function of many genes, proteins, and enzymes implicated in cell-cycle progression, resulting in dysregulated proliferation. Modulation of DNA replication, cell-cycle progression, and mitogenic signaling are meaningful cancer-cell targets for an anti-proliferative strategy [97]. Abundant research suggests the potential of curcumin for anti-proliferation in various types of cancer cells, whereas tetrahydrocurcumin has received little mention. It was reported that in the tumor-promoter TPA-treated mouse epidermis model, tetrahydrocurcumin was found to (slightly) reduce ornithine decarboxylase (ODC) activity, an enzyme involved in polyamine biosynthesis [69]. Polyamines are essential for cell growth and differentiation by maintaining nucleic acid and chromatin structure, regulating ion channels, and protein synthesis. Upregulation of ODC increases polyamine synthesis that, in turn, promotes tumor initiation and growth. Inhibition of ODC and polyamine synthesis is a rational approach for the suppression of abnormal proliferation in cancer cells [98]. 

Tetrahydrocurcumin decreased TPA-induced anchorage-independent growth in JB6 cells, indicating its ability to mitigate the progression of transformed cells to a tumor phenotype; however, the effect was markedly lower than that exhibited by curcumin [69]. In 1, 2-dimethylhydrazine dihydrochloride (DMH)-initiated B6C3F1 mice, post-administration of 0.5% tetrahydrocurcumin for 7 weeks significantly reduced the number of preneoplastic aberrant crypt foci (ACF) in the colon. The dietary tetrahydrocurcumin application was more effective than curcumin in reducing ACF formation and proliferation of the colon crypts, as monitored by BrdU labeling [70]. Moreover, dietary tetrahydrocurcumin administered for 23 weeks was more effective than curcumin in reducing azoxymethane (AOM)-induced colonic large-ACF formation in mice. Tetrahydrocurcumin was superior to curcumin in downregulating iNOS and COX-2 and decreasing proliferative signaling of Wnt-1/β-catenin/p-GSK3β in the colon [71]. These two studies suggest that dietary intake of tetrahydrocurcumin exerts a better in vivo biological effect on colonic carcinogenesis than curcumin. Recently, Zhang et al. reported that tetrahydrocurcumin enhanced the radio-sensitivity of glioma cells; acting synergistically when combined with radiation, it inhibited colony formation in the cells. However, the radio-sensitivity was much weaker than when radiation was combined with curcumin. The anti-proliferative effect of tetrahydrocurcumin decreased cyclin D1, a regulator of G1 phase progression of the cell cycle, and proliferating cell nuclear antigen (PCNA), a nuclear protein involved in DNA synthesis. Both of these interactions resulted in G0/G1 cell-cycle arrest in glioma cells. The combined treatment also exhibited stronger inhibitory effects on glioma growth in a xenograft mouse model [72]. Notably, this study demonstrated that tetrahydrocurcumin decreased intracellular glutathione (GSH) in glioma cells: elevated GSH levels have been shown to increase the anti-oxidative capacity of cancer cells against oxidative stress, which plays a key role in therapeutic resistance [99]. The effect of tetrahydrocurcumin alone or in combination with radiation on decreasing GSH content may be related to its anti-proliferative effect in glioma cells [72]. Because tetrahydrocurcumin is unable to react with thiols and deplete cellular GSH—due to the lack of unsaturated β-diketone group present in curcumin—it is not clear how the reduction of GSH is mediated; further examination of this is needed. Indeed, an early study revealed that tetrahydrocurcumin decreased GSH content in myelogenous leukemia cells without stimulating ROS production [39]. 

### 3.5. Anti-Metastatic Activity 

Cancer metastasis is a dynamic multi-step process that accounts for over 90% of cancer-related mortality. This process involves the spread of cancer cells from the original tissue into the lymphatic or vascular circulation and subsequent formation of secondary tumors in distant organs. Several cellular events have been shown to participate in cancer metastasis, such as loss of cell–cell adhesion, epithelial–mesenchymal transition (EMT), increased motility and migration, invasion of surrounding tissue, and angiogenesis. Interruption of these critical events provides therapeutic strategies for preventing cancer metastasis, disease progression, and consequent death [100]. Several studies have indicated that tetrahydrocurcumin appeared to be an excellent inhibitor of cancer cell metastasis. In an in vitro study, tetrahydrocurcumin inhibited the motility and invasion of highly metastatic HT1080 human fibrosarcoma cells in a trans-well assay. Increased levels of proteases such as matrix metalloproteinases (MMPs) and urokinase-type plasminogen activator (uPA) are involved in the degradation of extracellular matrix (ECM) during cell migration and invasion. This study showed that tetrahydrocurcumin not only abrogated MMP-2, MMP-9, and uPA activity but also reduced the protein expression of membrane-type 1-MMP in HT1080 cells. Moreover, tetrahydrocurcumin decreased the adhesion of HT1080 cells to Matrigel and laminin-coated plates but showed no inhibitory effect on adhesion to fibronectin and Type IV collagen-coated wells, providing further support for the interfering effect of tetrahydrocurcumin on the interaction of ECM and cancer cells [73]. Yoysungnoen et al. reported that tetrahydrocurcumin displayed superior anti-angiogenic effects than curcumin in vivo. In BALB/c-nude mice implanted with HepG2 xenografic tumor, daily oral administration of 3000 mg/kg tetrahydrocurcumin markedly reduced capillary vascularity within the tumor tissue, exhibiting a better inhibitory effect than curcumin after 21 days of treatment. However, the mechanism by which tetrahydrocurcumin reduced the density of neo-capillaries in tumor tissue was not further examined [46]. Similar tetrahydrocurcumin anti-angiogenic efficacy was also observed in cervical-tumor tissue [74]. BALB/c-nude mice with implanted CaSki tumors that were orally administered 100, 300, and 500 mg/kg of tetrahydrocurcumin daily for 30 days showed a markedly reduced neo-capillary network and CD31 level within the tumor tissue. The rapid proliferation of cancer cells within tumor tissue creates a hypoxic condition that triggers the angiogenic program to upregulate hypoxia-inducible factor (HIF)-1α and downstream vascular endothelial growth factor (VEGF), leading to tumor angiogenesis and tumor progression. Oral administration of tetrahydrocurcumin significantly decreased the levels of HIF-1α, VEGF, and VEGF receptor-2 in the CaSki tumor tissue; this supports the inhibitory effect of tetrahydrocurcumin on tumor angiogenesis [74]. The study also found that tetrahydrocurcumin downregulated COX-2, EGFR, p-ERK1/2, and p-AKT in CaSki tumor tissue; it is likely that this downregulation is a part of the overall mechanism of angiogenesis inhibition and tumor-volume reduction [75]. 

In a recent study, Zhang et al. reported the in vivo anti-metastasis efficacy of tetrahydrocurcumin. In the lung-metastasis model generated by injecting osteosarcoma cells into the tail veins of nude mice, intraperitoneal injection of 100 mg/kg tetrahydrocurcumin (5 days/week) for 8 weeks significantly reduced the number of metastatic nodules. The animals showed no weight loss and no incidence of acute or delayed toxicity [76]. The molecular mechanism underlying the anti-metastatic properties of tetrahydrocurcumin was explored in an in vitro study utilizing osteosarcoma cells. Tetrahydrocurcumin treatment upregulated epithelial E-cadherin and downregulated N-cadherin, vimentin, Snail, ZEB, and Twist mesenchymal markers associated with the mesenchymal-epithelial transition (MET) process. Moreover, tetrahydrocurcumin suppressed HIF-1α and its downstream targets such as VEGF, MMP2, and MMP9 (via interfering with Akt/mTOR and p38 MAPK signaling) that are involved in invasion and angiogenesis [76]. More importantly, tetrahydrocurcumin directly inhibited cell proliferation and HIF-1α accumulation, upregulated VEGF, and induced the MET process in osteosarcoma cells under hypoxic conditions. Hypoxia-induced capillary tube-like structures in human umbilical-vein endothelial cells were also mitigated by tetrahydrocurcumin. This study indicated that the effect on HIF-1α might make tetrahydrocurcumin a potential candidate for preventing cancer metastasis [76]. 

### 3.6. Induction of Programmed Cell Death 

Apoptosis, a type of programmed cell death, is a pivotal physiological process involved in embryonic development, cell differentiation, and maintenance of cellular homeostasis within tissues. This process is tightly controlled by a range of extracellular and intracellular signaling cascades, gene transcription, molecule assembly, and activation of effector caspases that cleave various proteins as a part of morphological and biochemical characteristics of apoptosis. A defect in any step of apoptosis leads to malignant transformation of initiated cells, abnormal proliferation, and resistance to chemotherapeutic agents [101]. Induction of apoptosis is a promising and popular strategy for current cancer therapy. In addition, an effective approach for overcoming cancer resistance to chemotherapy by enhancement of apoptosis or other mechanisms is urgently needed [101]. 

It is not surprising that the use of tetrahydrocurcumin is rarely considered since it is less toxic than curcumin [44,45,46]. Nonetheless, several studies reported the effect of tetrahydrocurcumin on inducing apoptosis in cancer cells. Kang et al. showed that tetrahydrocurcumin exhibited a cytotoxic effect on human breast cancer cells with IC_50_ of 33.08 μM, compared to 19.68 μM for curcumin [77]. Tetrahydrocurcumin induced mitochondria-dependent apoptosis in MCF-7 cells, evidenced by the loss of mitochondrial transmembrane potential (ΔΨm), upregulated Bax, decreased Bcl-2, and the release of cytochrome c as well as the cleavage of procaspases. Upregulation of p21, a cyclin-dependent kinase inhibitor, is also involved in tetrahydrocurcumin-triggered apoptosis [77]. A similar result was observed in a study conducted by Han et al., i.e., tetrahydrocurcumin-induced mitochondria-dependent apoptosis may be attributed to the elevation of intracellular ROS in MCF-7 cells [78]. In an H22 ascites tumor-bearing model, intraperitoneal injection in mice of 5, 10, and 20 mg/kg tetrahydrocurcumin daily for 7 days significantly increased the survival rates over 100 mg/kg curcumin treatment. A greater decrease in tumor weight and ascites volume was seen using 20 mg/kg tetrahydrocurcumin than a higher dose of curcumin. In addition, tetrahydrocurcumin was more effective in increasing apoptosis in H22-induced ascites tumors than curcumin. At the molecular level, both tetrahydrocurcumin and curcumin triggered a mitochondrial-dependent cascade, upregulation of p53, and decreased MDM2 (a negative regulator of p53) to induce apoptosis [79]. Although they share the same molecular mechanism, tetrahydrocurcumin exerted superior in vivo anti-tumor and apoptosis-inducing effects to those of curcumin. Two studies suggested a sensitizing effect of tetrahydrocurcumin on the chemotherapy and radiotherapy of malignant glioma. Pretreatment with 20 μM tetrahydrocurcumin markedly enhanced radiation-induced apoptosis in glioma C6 cells [72]. Using tetrahydrocurcumin and doxorubicin-loaded transferrin-conjugated nanoparticles caused a significantly larger cytotoxic effect than treatment with doxorubicin-loaded nanoparticles alone in glioma cells. Pretreatment with tetrahydrocurcumin and doxorubicin-loaded transferrin-modified nanoparticles sensitized glioma C6 cells to radiation, demonstrated by decreased colony formation [102]. These studies indicated that the combination of tetrahydrocurcumin is a promising strategy for sensitizing radiotherapy of glioma. However, the specific molecular mechanism responsible for the tetrahydrocurcumin-sensitizing effect on radiation was not investigated. 

Autophagy is another type of programmed cell death. Although its role in cancer is complicated, it may present an effective therapeutic strategy for cancer treatment [103]. Autophagy appears to be a specific cytotoxicity activity of tetrahydrocurcumin that is completely different from the activity of curcumin on cancer cells. Wu et al. suggested that tetrahydrocurcumin caused the formation of acidic vascular organelles (AVOs), whereas curcumin induced apoptosis in human HL-60 promyelocytic leukemia cells. Upregulation of the autophagosomal marker LC3 reflected tetrahydrocurcumin-induced autophagic cell death. They also showed that tetrahydrocurcumin triggered autophagy in HL-60 cells by impairing PI3K/Akt and mTOR/p70S6K signaling [80]. In cytarabine chemotherapy-resistant HL-60 cells, tetrahydrocurcumin induced autophagy by upregulating p62 and LC3, whereas curcumin caused apoptosis via activation of caspases [81]. This information suggests a novel therapeutic application of tetrahydrocurcumin for overcoming drug resistance in cancers. Tetrahydrocurcumin inhibited the efflux function of P-glycoprotein as well as ABCC1 and ABCG2 (both multidrug resistance proteins) and thus increased the sensitivity of drug-resistant cancer cells to chemotherapeutic drugs. The mechanism by which tetrahydrocurcumin overcomes the reverse multidrug resistance phenotype might be related to its interaction with the drug-binding site of the transporter; this was proved by the increased ATPase activity of P-glycoprotein and ABCG2 [104]. The autophagy-inducing effect of tetrahydrocurcumin was also found in non-small-cell lung carcinoma cells. Treatment of A549 lung cancer cells with tetrahydrocurcumin resulted in autophagic cell death through an increase in the LC3 I/II ratio and Beclin-1 while in parallel downregulating PI3K/Akt/mTOR signaling [82]. 

### 3.7. Immuno-Modulating Activity 

Cancer cells evade immune surveillance and destruction through multiple mechanisms. Disruption of phagocytic clearance in macrophages and myeloid immune cells is an important mechanism for cancer invasion and metastasis. It is known that cancer cells express the CD47 receptor to manipulate macrophages that escape immune surveillance; blockage of the CD47/signal regulatory protein alpha (SIRPα) axis increases phagocytosis of cancer cells and immune responses [105]. Two in vitro studies showed that tetrahydrocurcumin stimulated phagocytic activity in RAW264.7 macrophages with or without LPS stimulation [83,84]. In addition, tetrahydrocurcumin enhanced LPS-stimulated NK-cell activity in culture supernatants (mouse splenocytes). The stimulatory effect of tetrahydrocurcumin was more potent than that of curcumin [84]. However, the mechanism by which tetrahydrocurcumin stimulates phagocytosis needs further investigation. 

## 4. Enhancement of Tetrahydrocurcumin Biological Activity Via Structural Modification and Increased Delivery

In the absence of the α, β-unsaturated β-diketo moiety, tetrahydrocurcumin is less reactive than curcumin in nucleophilic-addition reactions [19,36]. Structural modification of tetrahydrocurcumin may improve the original anti-cancer activity and generate additional, distinct biological functions. A recent study showed that a synthetic tetrahydrocurcumin-iridium (Ir)^III^ complex demonstrated a remarkable and rapid cytotoxic effect in HeLa cells when irradiated with visible light. The phototoxic effect triggered by tetrahydrocurcumin-Ir ^III^ was higher than that of curcumin-Ir ^III^, providing a novel application in photodynamic therapy [106]. Mahal et al. synthesized pyrazole- and Schiff-base derivatives of tetrahydrocurcumin by its direct condensation with various hydrazines and primary amines, respectively. Most of these synthetic pyrazole- and Schiff-base tetrahydrocurcumin derivatives exhibited more potent cytotoxic effects than tetrahydrocurcumin in A549, HeLa, and MCF-7 cancer cells [107,108]. 

Even though tetrahydrocurcumin is believed to have a superior biological effect to curcumin in vivo, its use is still limited by low water solubility. Different nanotechnologies are showed to improve aqueous solubility of tetrahydrocurcumin. Two studies reported a self-emulsifying drug delivery system by mixture of different oil, surfactant, and cosurfactant to form tetrahydrocurcumin nanoemulsions, which significantly elevated their dissolution in simulated gastric fluid compared to tetrahydrocurcumin unmodified [109,110]. Kakkar et al. prepared lipid nanoparticles of tetrahydrocurcumin which showed a significantly higher and fast drug release than free tetrahydrocurcumin in a media mixed with alcohol and phosphate buffer. The increased drug release of tetrahydrocurcumin is due to the enhancement of aqueous solubility of tetrahydrocurcumin [111]. Other studies have shown that conjugation of tetrahydrocurcumin with a glucosyl group or formulation with carboxymethylcellulose by micro emulsification or liquid self-emulsifying techniques elevate its bioavailability. Using the above approaches, the delivery, absorption, and controllable release of tetrahydrocurcumin were significantly improved and resulted in the enhancement of its biological efficacy [110,111,112,113]. The above studies suggest that tetrahydrocurcumin nanoparticles and conjugates may in the future act as promising chemo-preventive and therapeutic agents; however, the pharmacokinetics of such constructs are yet to be determined.

## 5. Conclusions and Future Perspectives

Tetrahydrocurcumin is considered to be a valuable lead compound for developing cancer prevention and treatment therapeutics due to its relatively superior chemical stability and bioavailability, as well as its high structural similarity to curcumin. The anti-cancer properties of tetrahydrocurcumin discussed in this review are proposed to be due to its strong anti-oxidative capability. They are generated via regulation of multiple signaling molecules and gene expression, interaction with enzymes and proteins implicated in inflammation, proliferation, metastasis, programmed cell death, and immune function (Figure 3). Although tetrahydrocurcumin lacks the reactive α, β-unsaturated β-diketo moiety, it exists in different keto-enol forms and other analogs that may be associated with the functional and mechanistic diversity that is different from that of curcumin. Prospective studies to identify the exact molecular targets of tetrahydrocurcumin will guide this opportunity towards the development of novel and more effective preventive and therapeutic approaches against various human malignancies. 

Despite the beneficial anti-cancer effects of tetrahydrocurcumin, many challenges must be overcome for its clinical translation. For example, tetrahydrocurcumin displayed more potent beneficial effects than curcumin in several studies; however, most of them were based on cell line and animal studies. The clinical studies for the promising anti-cancer potential of tetrahydrocurcumin and investigation of its exact mechanism of action are needed. In addition, the poor bioavailability of tetrahydrocurcumin is a major limitation for its clinical application. Although several delivery systems have been developed to improve the aqueous solubility, uptake, and drug release of tetrahydrocurcumin, it is required to confirm these in clinical trials. Examination of the physiologically relevant dose of tetrahydrocurcumin for its clinical safety and efficiency on human is also important. 

## Figures and Tables

**Figure 1 biomolecules-10-00831-f001:**
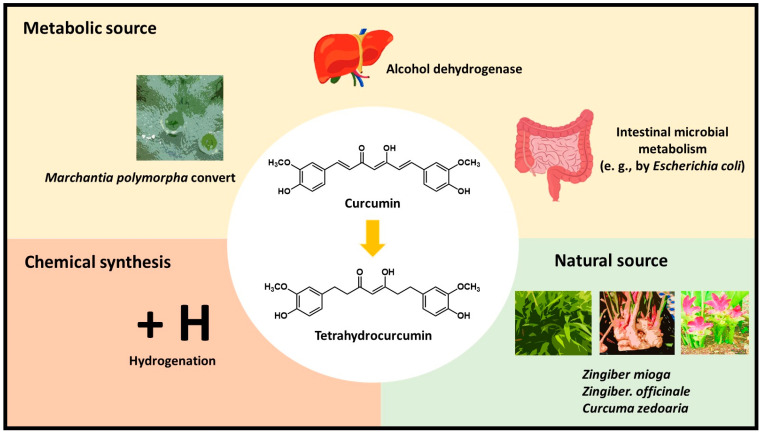
The sources of tetrahydrocurcumin.

**Figure 2 biomolecules-10-00831-f002:**
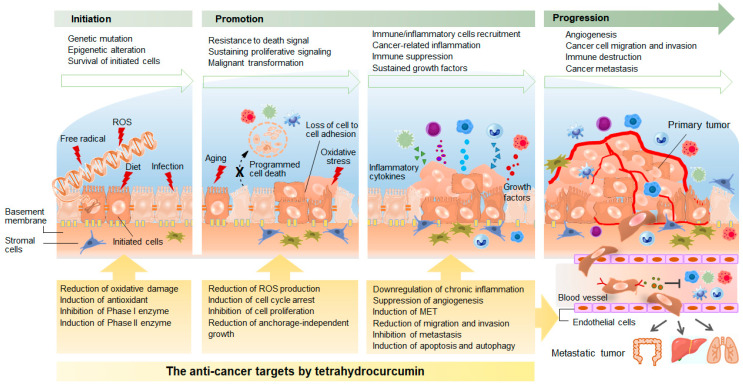
Mechanisms underlying multi-step tumorigenesis and the mechanism by which tetrahydrocurcumin exerts its anticancer action. ROS, reactive oxygen species; MET, mesenchymal-epithelial transition

**Figure 3 biomolecules-10-00831-f003:**
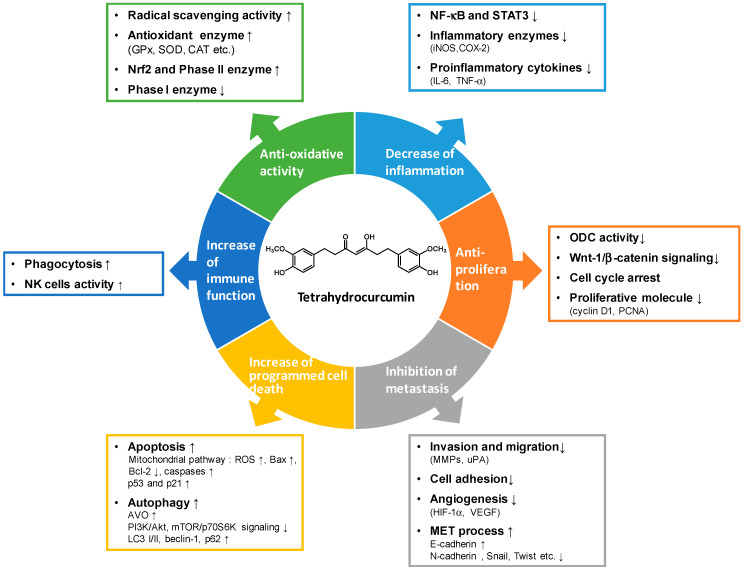
The potential mechanism of actions and molecular targets for tetrahydrocurcumin in prevention and treatment of cancer.

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
