# Peer review of "The Cancer Chemopreventive and Therapeutic Potential of Tetrahydrocurcumin"

_biomolecules, 2020, doi:10.3390/biom10060831_

Round 1

Reviewer 1 Report

The review article entitled "The cancer chemopreventive and therapeutic
potential of tetrahydrocurcumin" by Lai et al. is nicely written with enough information. This review article is suitable for publication in the Biomolecules Journal with some minor changes.

  1. In figure 1, structure of tetrahydrocurcumin was wrong, please correct the structure.
  2. Add some more appropriate figures to show its effectiveness for cancer therapy.
  3. Tetrahydrocurcumin is sparingly soluble in aqueous buffers, please add something about nanoformulation to improve its water solubility.
  4. Add future perspective

Author Response

## Reviewer 1 Comments                                               

Major concern

  1. In figure 1, structure of tetrahydrocurcumin was wrong, please correct the structure.

Ans: Thank you for your correction. We have corrected the structure of tetrahydrocurcumin in Figure 1 in page 4.

  1. Add some more appropriate figures to show its effectiveness for cancer therapy.

Ans: Thank you for your comment. We have added Figure 3 to show the potential mechanism of actions and molecular targets for tetrahydrocurcumin in prevention and treatment of cancer in page 20.

  1. Tetrahydrocurcumin is sparingly soluble in aqueous buffers, please add something about nanoformulation to improve its water solubility.

Ans: Thank you very much for your comment. We have included several studies of tetrahydrocurcumin nanoformulation related to improve its aqueous solubility in page 20, Line 630-638, as following,

Different nanotechnologies are showed to improve aqueous solubility of tetrahydrocurcumin. Two studies reported a self-emulsifying drug delivery system by mixture of different oil, surfactant, and cosurfactant to form tetrahydrocurcumin nanoemulsions, which significantly elevated their dissolution in simulated gastric fluid compared to tetrahydrocurcumin unmodified [109,110]. Kakkar et al prepared lipid nanoparticles of tetrahydrocurcumin which showed a significantly higher and fast drug release than free tetrahydrocurcumin in a media mixed with alcohol and phosphate buffer. The increased drug release of tetrahydrocurcumin is due to the enhancement of aqueous solubility of tetrahydrocurcumin [111].

Refs:

  1. Astuti, I. Y., Suliatin, T., and Wahyuningrum, R. Dissolution enhancement of tetrahydrocurcumin using optimized self-nanoemulsifying drug delivery system. Int J App Pharm 2019, 11, 97-102.
  2. Setthacheewakul, S., Kedjinda, W., Maneenuan, D., and Wiwattanapatapee, R. Controlled release of oral tetrahydrocurcumin from a novel self-emulsifying floating drug delivery system (SEFDDS). AAPS.PharmSciTech. 2011, 12, 152-164.
  3. Kakkar, V., Kaur, I. P., Kaur, A. P., Saini, K. et al., Topical delivery of tetrahydrocurcumin lipid nanoparticles effectively inhibits skin inflammation: in vitro and in vivo study. Drug Dev.Ind.Pharm 2018, 44, 1701-1712.

  1. Add future perspective.

Ans: Thank you very much for your comment. We have added future perspective in page 21, Line 660-669, as following,

Despite the beneficial effects of tetrahydrocurcumin on anti-cancer, many challenges must be overcome for its clinical translation. For example, tetrahydrocurcumin displayed more potent beneficial effects than curcumin in several studies; however, most of them based on cell line and animal studies. The clinical studies for the promising anti-cancer potential of tetrahydrocurcumin and investigation of its exact mechanism of action are needed. In addition, the poor bioavailability of tetrahydrocurcumin is a major limitation for its clinical application. Although several delivery systems have been developed to improve the aqueous solubility, uptake and drug release of tetrahydrocurcumin while it is required to be confirmed in clinical trials. Examination of the physiologically relevant dose of tetrahydrocurcumin for its clinical safety and efficiency on human is also important.

Reviewer 2 Report

In this review article, Lai et al have discussed in detail the anticancer properties of Tetrahydrocurcumin, a curcumin metabolite. The clinical potential of curcumin is limited by its poor bioavailability. Therefore, the researchers have long been trying to overcome these limitations of curcumin by synthetic analogs and other innovative strategies. Tetrahyrocurcumin has shown improved oral bioavailability and water solubility than curcumin, thus potentially making it better clinical candidate than curcumin.  Although there are many review articles on the anticancer properties of curcumin, but a comprehensive review article like this on cancer chemopreventive and therapeutic effects of Tetrahydrocurcumin is lacking. The review is organized and well written. I have few comments:

  1. While the authors have discussed in detail how tetrahydrocurcumin is superior to curcumin in terms of better water solubility, oral bioavailability and therefore improved anticancer activities, but they have not touched upon the relative efficacy of tetrahydrocurcumin compared to synthetic analogs of curcumin in light of above parameters. So, it would be nice if authors can devote a paragraph on this.
  2. There are several instances where authors have mentioned about certain studies without proper citation. For example, authors should cite the references after “…………a comprehensive cellular and animal study” in line number 62, and after “………. treatment of various cancers” in line number 91. There are also no reference citations in line 328-344. Please check all such instances throughout the manuscript and cite the appropriate references.
  3. Should it be ‘interfering’ in place of “interferon” in line 513?

Author Response

## Reviewer 2 Comments                                              

  1. While the authors have discussed in detail how tetrahydrocurcumin is superior to curcumin in terms of better water solubility, oral bioavailability and therefore improved anticancer activities, but they have not touched upon the relative efficacy of tetrahydrocurcumin compared to synthetic analogs of curcumin in light of above parameters. So, it would be nice if authors can devote a paragraph on this.

Ans: Thank you very much for your comment. Extensive research conducted to designed and synthesized curcumin analogs within past years and many of them are found to with improved chemical stability, solubility and biological activity in comparison with curcumin; however, very few studies were performed to compare the bioavailability or biological activity between synthetic curcumin analogs and tetrahydrocurcumin. We have included two comparative studies of dimethoxycurcumin and tetrahydrocurcumin for their anti-inflammatory activity in page 15, Line 400-411, as following,

Many synthetic analogs of curcumin have developed by modification of its structural motifs to exhibit greater chemical stability, bioavailability and biological activities, and has been discussed in several review articles [15,90,91]. Some of them displayed a better aqueous solubility, cellular uptake, in vivo bioavailability and anti-cancer effect than original curcumin [92-94]; however, relatively few comparative studies were conducted between curcumin analogs and tetrahydrocurcumin. Pae et al. reported the dimethoxycurcumin, a synthetic analog with higher metabolic stability over curcumin, was more potent than curcumin and tetrahydrocurcumin to suppress NO production in LPS-stimulated RAW264.7 murine macrophages [95]. Their latter study also showed the synthetic dimethoxycurcumin was more effective than tetrahydrocurcumin on induction of Nrf-2 dependent HO-1 expression in RAW264.7 murine macrophages[96]. These studies indicated not only the two conjugated double bonds in curcumin are required for anti-inflammatory activity but also the increased number of methoxy groups.

Refs:

  1. Vyas, A., Dandawate, P., Padhye, S., Ahmad, A. et al., Perspectives on new synthetic curcumin analogs and their potential anticancer properties. Curr.Pharm Des 2013, 19, 2047-2069.
  2. Padhye, S., Chavan, D., Pandey, S., Deshpande, J. et al., Perspectives on chemopreventive and therapeutic potential of curcumin analogs in medicinal chemistry. Mini.Rev.Med Chem. 2010, 10, 372-387.
  3. Bairwa, K., Grover, J., Kania, M., and Jachak, S. M. Recent developments in chemistry and biology of curcumin analogues. RSC Adv. 2014, 4, 13946-13978.
  4. Liang, G., Shao, L., Wang, Y., Zhao, C. et al., Exploration and synthesis of curcumin analogues with improved structural stability both in vitro and in vivo as cytotoxic agents. Bioorg.Med Chem 2009, 17, 2623-2631.
  5. Khudhayer Oglah, M. and Fakri Mustafa, Y. Curcumin analogs: synthesis and biological activities. Med Chem Res 2020, 29, 479-486.
  6. Ramayanti, O., Brinkkemper, M., Verkuijlen, S. A. W. M., Ritmaleni, L. et al., Curcuminoids as EBV Lytic Activators for Adjuvant Treatment in EBV-Positive Carcinomas. Cancers.(Basel) 2018, 10.
  7. Pae, H. O., Jeong, S. O., Kim, H. S., Kim, S. H. et al., Dimethoxycurcumin, a synthetic curcumin analogue with higher metabolic stability, inhibits NO production, inducible NO synthase expression and NF-kappaB activation in RAW264.7 macrophages activated with LPS. Mol.Nutr.Food Res 2008, 52, 1082-1091.
  8. Jeong, S. O., Oh, G. S., Ha, H. Y., Soon, K. B. et al., Dimethoxycurcumin, a Synthetic Curcumin Analogue, Induces Heme Oxygenase-1 Expression through Nrf2 Activation in RAW264.7 Macrophages. J Clin Biochem Nutr. 2009, 44, 79-84.

  1. There are several instances where authors have mentioned about certain studies without proper citation. For example, authors should cite the references after “…………a comprehensive cellular and animal study” in line number 62, and after “………. treatment of various cancers” in line number 91. There are also no reference citations in line 328-344. Please check all such instances throughout the manuscript and cite the appropriate references.

Ans: Thank you for your comment. We have carefully checked throughout the manuscript and included appropriate references which marked by red color.

  1. Should it be ‘interfering’ in place of “interferon” in line 513?

Ans: Thank you for your comment. We have corrected the “interferon” to “interfering” in page 17, Line 530.